# Can the Addition of Maintenance Electroconvulsive Therapy to Pharmacotherapy Improve Relapse Prevention in Severe Major Depressive Disorder? A Randomized Controlled Trial

**DOI:** 10.3390/brainsci11101340

**Published:** 2021-10-11

**Authors:** Erika Martínez-Amorós, Narcís Cardoner, Verònica Gálvez, Aida de Arriba-Arnau, Virginia Soria, Diego J. Palao, José M. Menchón, Mikel Urretavizcaya

**Affiliations:** 1Department of Mental Health, Parc Taulí University Hospital, Institut d’Investigació i Innovació Sanitària Parc Taulí (I3PT), 08208 Sabadell, Spain; emartineza@tauli.cat (E.M.-A.); vgalvez@tauli.cat (V.G.); DPalao@tauli.cat (D.J.P.); 2Department of Psychiatry and Forensic Medicine, Universitat Autònoma de Barcelona, 08193 Bellaterra, Spain; 3Centro de Investigación Biomédica en Red de Salud Mental (CIBERSAM), Carlos III Health Institute, 28029 Madrid, Spain; vsoria@bellvitgehospital.cat (V.S.); jmenchon@bellvitgehospital.cat (J.M.M.); 4Department of Psychiatry, Bellvitge University Hospital, Bellvitge Biomedical Research Institute (IDIBELL), Neurosciences Group—Psychiatry and Mental Health, 08907 L’Hospitalet de Llobregat, Spain; adearriba@bellvitgehospital.cat; 5Department of Clinical Sciences, School of Medicine, Universitat de Barcelona, 08007 Barcelona, Spain

**Keywords:** electroconvulsive therapy, major depressive disorder, prevention, relapse, recurrence, maintenance

## Abstract

Few systematic evaluations have been performed of the efficacy of electroconvulsive therapy (ECT) as a relapse prevention strategy in major depressive disorder (MDD). This is a single-blind, multicenter, randomized controlled trial to compare the efficacy and tolerability of pharmacotherapy plus maintenance ECT (M-Pharm/ECT) versus pharmacotherapy alone (M-Pharm) in the prevention of MDD relapse. Subjects with MDD who had remitted with bilateral acute ECT (n = 37) were randomly assigned to receive M-Pharm/ECT (n = 19, 14 treatments) or M-Pharm (n = 18) for nine months. The subjects were followed up for 15 months. The main outcome was relapse of depression, defined as a score of 18 or more on the Hamilton Depression Rating Scale. At nine months, 35% of the subjects treated with M-Pharm/ECT relapsed as compared with 61% of the patients treated with M-Pharm. No statistically significant differences between groups were indicated by either Kaplan–Meier or Cox proportional hazards regression analyses. The subjects without psychotic features were at higher risk of relapse. There were no statistically significant differences in the MMSE scores of the two groups at the end of the study. Further studies are needed to better define the indications for M-ECT in order to improve its efficacy as a relapse prevention strategy.

## 1. Introduction

Electroconvulsive therapy (ECT) is an effective acute treatment in severe forms of depression [1,2,3]. However, subjects with major depressive disorder (MDD) who have responded to a course of ECT during the acute phase show a high risk of relapse, despite the use of appropriate continuation or maintenance strategies [4]. In fact, nearly 40% of patients who respond to ECT are expected to relapse within the first six months, and approximately 50% of patients by the end of the first year [4]. 

Since the introduction of ECT, the possibility of continuing its application after recovery from an episode has been suggested, with the aim of prolonging improvement after an acute treatment course (maintenance ECT [5,6,7], prophylactic electroshock [8,9], or convulsion dependence [10]). In this regard, the American Psychiatric Association (APA) considers ECT to be a viable therapeutic option for long-term management of subjects with MDD who required ECT in the acute phase [1]. While the term “continuation ECT” (C-ECT) is often used when the therapy is applied for prevention of relapse in the same depressive episode (i.e., within six months of remission), maintenance ECT (M-ECT) is normally used for recurrence prevention (i.e., prevention of a new episode, beyond these six months). In this paper, however, the term “maintenance ECT” (M-ECT) is used to refer to treatment to prevent both relapse and recurrence.

The randomized trials [11,12,13,14] and reviews [15,16,17,18,19,20,21,22,23] carried out in the last years have concluded that M-ECT may be a good long-term therapeutic alternative in MDD, effective in preventing relapse and recurrence, and able to reduce the number and duration of hospital admissions. Positive cost-effectiveness studies also support M-ECT programs [24,25,26]. Nonetheless, the indication of M-ECT has been questioned by some authors, who stress that the data available on its long-term efficacy and safety are scarce [27]. 

It is well established that ECT is a safe, well-tolerated strategy. Its most common acute adverse effects are headache, nausea, myalgia, dental damage, confusion, and cognitive effects [1,28]. Most of these adverse effects are self-limiting and can be managed symptomatically. Serious side effects are very uncommon and include cardiovascular, pulmonary, and cerebrovascular events [28]. The mortality related to ECT has been estimated to be 2.1 per 100,000 treatments—similar to, or even lower than, the rates in other surgical procedures involving general anesthesia [29]. ECT has been associated with cognitive disturbances, especially memory impairment [14,28]. Most cognitive adverse effects with ECT are short term, but retrograde amnesia, though infrequent, may persist [28]. In this regard, the data derived from randomized trials have suggested that M-ECT is not associated with long-lasting cognitive adverse effects [11,12,14,30].

The primary aim of this study was to assess the efficacy of M-ECT in MDD. The secondary aims were to assess the residual symptoms and the safety and tolerability (adverse effects and global cognitive functioning) of both maintenance treatments.

## 2. Materials and Methods

### 2.1. Study Design and Recruitment

A pragmatic, single-blind, multicenter, two-arm randomized trial was conducted (EudraCT number 2007-007166-37). The trial was registered at www.clinicaltrials.gov (NCT01305707; 1 March 2011). Subjects meeting the inclusion criteria were consecutively recruited at two Spanish university hospitals (Bellvitge University Hospital and Parc Taulí University Hospital), between 30 July 2009 and 31 December 2014.

The subjects were randomly assigned to one of the following two groups:(1)M-Pharm/ECT group, consisting of pharmacotherapy plus ECT for nine months;(2)M-Pharm group, pharmacotherapy alone.

In both groups, the pharmacotherapy used was the same as in the acute phase, without dose changes. The randomization was performed through a list of permutations.

Subjects were followed for nine months during the M-ECT treatment period, and then for a further six months, up to a total of 15 months. A total of 16 assessments were conducted at week 0 (baseline), weekly (assessments 1–4), every two weeks (assessments 5–8), and monthly (assessments 9–14). The last two assessments (15 and 16) were carried out at intervals of three months. Additional visits were performed when deemed clinically necessary, if the patient experienced a relapse or recurrence.

### 2.2. Subjects

All subjects were adults, and the following inclusion criteria were applied: (1) fulfillment of DSM-IV-TR [31] diagnostic criteria for unipolar MDD, who required ECT during the index episode (i.e., the current episode prior to randomization treated with acute ECT), subjects’ suitability for ECT was determined on a clinical basis, and the acute ECT course was applied in accordance with the APA Task Force on ECT [1] and the Spanish Consensus on ECT [32]; (2) the achievement of full clinical remission with acute ECT, defined as a score of ≤7 on the 21-item Hamilton Depression Rating Scale (HDRS-21 [33,34]) on two consecutive assessments [35]; (3) the ability to provide informed consent; (4) in the case of women of childbearing age, the use of medically accepted methods of contraception.

Exclusion criteria were: (1) the presence of comorbid axis I or II disorders (except nicotine dependence); (2) prior inclusion on the M-ECT program; (3) administration of ECT in the last three months; and (4) pregnancy or nursing.

Informed consent was obtained from all subjects. The study was conducted in accordance with the guidelines of the Declaration of Helsinki and was approved by the Bellvitge University Hospital Ethics Committee and the Parc Taulí University Hospital Ethics Committee.

### 2.3. Intervention/Treatment

#### 2.3.1. M-ECT

The following schedule of sessions was used in the M-Pharm/ECT group, based on the national ECT guidelines [32]: weekly ECT for four weeks (four sessions), every two weeks for two months (four sessions), and monthly for six months (six sessions), up to a total of 14 sessions in nine months. M-ECT was administered using a brief pulse constant-current device (Thymatron^TM^ DGx and System IV, Somatics, Inc., Lake Bluff, IL, USA), with bilateral (bifrontotemporal) electrode placement. General anesthesia was induced with intravenous thiopental (2–2.5 mg/kg) or propofol (0.75–1 mg/kg), and succinylcholine (0.5–1 mg/kg) was used as a muscle relaxant. When subjects showed adequate muscle relaxation, they were preoxygenated, and then manually ventilated using a valve mask and 100% oxygen. At the beginning of M-ECT, the electrical dose administered was the same as at the end of the acute ECT. The mean (SD) electrical dosage during M-ECT was 258.0 (98.3) mC, with a pulse width of 1 ms in all subjects. The mean EEG seizure duration was 37.1 (7.9) seconds.

#### 2.3.2. Pharmacotherapy

Inclusion in the study did not affect the choice of pharmacological treatment; in both groups, pharmacological treatment was prescribed by the treating psychiatrist during the acute ECT course, and this was continued during the follow-up, with no changes in the doses or type of drugs. Therefore, all subjects received individualized medication, and all medications were allowed. In the overall sample, the concomitant pharmacotherapy (Table 1) consisted of antidepressants (100%), antipsychotics (63.9%), benzodiazepines (69.4%), and lithium (2.8%). The M-Pharm/ECT group had a lower % of antipsychotic use at baseline (*p* = 0.035). No significant differences were found in the antidepressant strategy (monotherapy, combination and augmentation) between the M-Pharm/ECT group and the M-Pharm group (*p* = 0.231).

The treating psychiatrist was nonblinded in both groups. If the subject experienced a relapse or recurrence, any change in treatment, including ECT, was allowed regardless of the study treatment group, that is, all study subjects always received the most appropriate treatment based on their clinical status.

### 2.4. Variables and Measures

Data were collected on demographic characteristics, the course of the disorder, and the following clinical and therapeutic variables relative to the index episode and maintenance treatment: sex and age; age at onset; number of episodes; concomitant medication; data regarding the acute ECT course and M-ECT, including the number of ECT sessions administered in the index episode; and ECT parameters such as pulse-width, duration of seizure, and charge.

The study protocol was administered by a blinded investigator. The primary instrument used to rate depressive symptoms was the HDRS-21. The study protocol at baseline also included: (i) the presence of melancholic, psychotic, and catatonic symptoms using DSM-IV-TR major depression specifiers; (ii) psychomotor disturbance, measured by the CORE system [36,37]; (iii) treatment-resistant depression, assessed by the Thase and Rush staging method [38], subjects were considered to be resistant if they reached Thase and Rush stage 3 or higher [39]; (iv) functional status, assessed using the Spanish version of the Global Assessment of Function (GAF, [40]); (v) clinical improvement and severity according to the Clinical Global Impression scale (CGI, [41]), CGI Improvement scale (CGI-I), and CGI Severity scale (CGI-S); (vi) adherence to treatment according to the Simplified Medication Adherence Questionnaire (SMAQ, [42]); (vii) adverse effects, assessed by the Utvalg for Kliniske Undersogelser (UKU) scale [43]; and (viii) global cognitive functioning using the Spanish version of the Mini-Mental State Examination (MMSE, [44]) [45]. Neurocognitive performance was also assessed using an extensive test battery. The neuropsychological battery was administered by neuropsychological technicians who were blinded to the treatment condition. The full results of the neuropsychological functioning battery are reported elsewhere.

In assessments 1–16, HDRS-21_,_ CGI-S, SMAQ, and UKU were administered. At assessments 14 and 16, MMSE and the neurocognitive battery were also administered.

The primary outcome measures in both groups were the relapse rate at nine months, and the time to relapse. Relapse was defined as: (1) a score of 15–17 on the HDRS^-21^ at two consecutive ratings or (2) a score of 18 or higher at one rating.

### 2.5. Statistics

A descriptive analysis of the sample was conducted. Demographic and clinical differences at baseline (at randomization) were tested treatment groups by using either the chi-squared test or Fisher’s exact test (for qualitative variables), and a Student’s *t*-test or Mann–Whitney U test (for quantitative variables). Categorical variables are reported as absolute frequencies and percentages (%), and quantitative variables as means (standard deviations) or medians.

An intention-to-treat analysis was conducted. Data missing due to withdrawal from the study were compiled using the LOCF (last observation carried forward) procedure, that is, imputing the last value recorded for the main variable to the rest of subsequent time points for this variable. The rest of the variables were analyzed using the ADO (available data only) procedure, that is, without imputation of missing variables.

The primary time point for treatment comparison was nine months. The relapse rate was estimated for each group as frequency in terms of percentage (%). A survival analysis was conducted using the Kaplan–Meier method and the log-rank test for estimating relapse time. Treatment was considered the main factor. Cox proportional hazard models were conducted; first, a univariate Cox analysis was performed to identify possible contributing factors, and second, the variables that showed a significant association in the univariate analysis (*p* < 0.05) were entered into a multivariate Cox regression model, together with the main factor (treatment).

Finally, three complementary analyses were conducted: (1) determination of the relapse rate at 15 months (end of follow-up), as frequency in percentage (%), and time to relapse using the Kaplan–Meier method and the log-rank test; (2) assessment of the evolution of depressive symptoms (HDRS-21) and severity (CGI-S) using a general linear model (GLM) to analyze the effect of the within-subjects factors (time and interaction between time and treatment arm) and the between-subjects factor (treatment arm); (3) analysis of possible differences between treatment groups with regard to adverse effects (UKU) and cognitive function (MMSE), using a Student’s *t*-test or repeated measures *t*-test, as appropriate.

All data were analyzed using SPSS for Windows, v. 19.0 or higher (SPSS, Chicago). Statistical significance was defined as *p* < 0.05 (two-sided).

### 2.6. Sample Size

Assuming an alpha risk of 0.05 and a beta risk of 0.20 in a one-sided contrast, and based on the primary efficacy variable, it was considered that 54 subjects in the first group and 54 subjects in the second group were required in order to detect, according to the best reported estimators, a difference equal to or greater than 0.25, assuming a ratio of 0.67 in the M-Pharm group [46]. A follow-up dropout rate of 0.1 was estimated.

### 2.7. Dropout from Statistical Analysis

Subjects were withdrawn from the study in the following cases: (a) the subject withdrew his/her consent; (b) the investigator considered that the best option for the subject was to abandon the study, for both safety and efficacy reasons; (c) pregnancy; and (d) loss to follow-up. In all cases of drop-out, the date and reason were noted. Subjects who withdrew from the study were followed up by their physician in charge. All those who initiated randomized treatment and returned for at least one post-baseline assessment were included in an intention-to-treat analysis.

## 3. Results

### 3.1. Subject Disposition

A total of 37 subjects were enrolled in the study: 22 from Bellvitge University Hospital (59.5%) and 15 (40.5%) from Parc Taulí University Hospital. Nineteen subjects (51.4%) were randomly assigned to the M-Pharm/ECT group (pharmacotherapy plus ECT for 9 months) and 18 subjects (48.6%) to the M-Pharm group (pharmacotherapy alone). The subject flowchart is shown in Figure 1. One subject withdrew informed consent at the baseline visit, after the randomization (analyzable for safety sample: n = 36); another subject was excluded for presenting a hip fracture in the first week of study, with no post-baseline assessments. Finally, 35 subjects (94.6% of those recruited) were considered analyzable for efficacy (ITT). Among these 35 subjects, three (8.6%) dropped out early: one subject in the M-Pharm group (due to psychiatric comorbidity, after Assessment 7, during the first nine months) and two subjects in the M-Pharm/ECT group (due to withdrawal of consent and an adverse effect, respectively, both after Assessment 14).

### 3.2. Study Sample Characteristics

The baseline characteristics of the subjects analyzable for safety are presented in Table 1 (at randomization); 66.7% were female and the median age was 67.5 years. As regards clinical characteristics, 97.2% of the subjects had melancholic features, and 58.3% of the subjects had psychotic features. The median duration of current episode was 0.5 (0.1–4.1) years. The mean age at illness onset was 50.4 ± 16.4 years, subjects had a median of three prior episodes of MDD (1–12), and 33.3% of subjects had received ECT in previous episodes. In the current episode, 96.8% of subjects had previously received pharmacological treatment; 44.4% of them had a high level of resistance to pharmacotherapy (median Thase and Rush stage 2 (0–5)). At baseline, the median HDRS-21 score was 2.5 (0–6). The subjects in the M-Pharm/ECT group had better scores on the CGI-I scale, with a median of 2 (much improved) vs. 3 (minimally improved) in the M-Pharm group, *p* = 0.017), and for neurological side effects on the UKU scale (with a mean ± SD UKU subscale score of 0.1 ± 0.2 as compared with 0.4 ± 0.8 in the M-Pharm group, *p* = 0.036). At baseline (the time of randomization), no other statistically significant differences in demographic or clinical variables were observed regarding treatment groups (Table 2).

### 3.3. Efficacy Results

#### 3.3.1. Relapse in the Intention-to-Treat Sample at Nine Months

The relapse rate at nine months was 48.6% (17/35 subjects). In the M-Pharm/ECT group, 35.3% relapsed (6/17), and 64.7% (11/17) remained in remission. In the M-Pharm group, 61.1% (11/18) relapsed, 33.3% (6/18) remained in remission, and 5.6% (1/18) dropped out early; the differences between groups were not statistically significant (*p* = 0.127). Nevertheless, the number needed to treat (NNT) was 3.87, meaning that it would be necessary to treat four subjects with pharmacotherapy as compared with M-ECT in order to avoid one relapse. The mean ± SD time to relapse was 3.1 ± 1.9 months among subjects who relapsed at nine months (n = 17), with no differences between treatment groups (3.4 ± 1.8 months in the M-Pharm/ECT group vs. 2.9 ± 2.1 months in the M-Pharm group, *p* = 0.615). The median (min-max) time to relapse was 3.1 months (0.5–6.6 months): 3 months (0.5–6.6) in the M-Pharm/ECT group and 3.3 months (0.8–5.6) in the M-Pharm group, indicating that 50% of relapses occurred within the first three months in both groups.

In the Kaplan–Meier survival analysis, the patterns of relapse-free times were similar in the two groups (log-rank test = 0.093, Figure 2).

In the univariate Cox analysis (Table 3), there were no statistically significant differences between the treatment groups (*p* = 0.102, the M-Pharm group was 2.3 times more likely to relapse than the M-Pharm/ECT group, 95% CI 0.85–6.23). Both the absence of psychotic symptoms (HR 3.54, 95% CI 1.29–9.70, *p* = 0.014) and age (≤67 years) influenced relapse (HR 3.46, 95% CI 1.21–9.88, *p* = 0.021). None of the other variables evaluated (sex, use of ECT in the past, episode duration, treatment-resistance, use of antipsychotics, center, or clinical features assessed by rating scales) appeared to influence relapse. In the multivariate Cox regression model (adjusted model, Table 3), only the absence of psychotic symptoms seemed to influence relapse.

To further investigate the reason for the significant differences between treatments in the model adjusted for psychotic symptoms (Model 2) we sought a possible relationship between the two variables. We found that the group of subjects with psychotic symptoms who underwent M-ECT (n = 8) presented zero events during the first nine months; a fact that determines statistically significant differences between both groups (ECT with psychotic symptoms vs. ECT without psychotic symptoms vs. pharmacotherapy with psychotic symptoms vs. pharmacotherapy without psychotic symptoms, log-rank test = 0.008). It can be concluded that the presence of psychotic symptoms influences recurrence-free survival, regardless of the treatment received by the subjects. Although subjects who received M-ECT (n = 8) showed a better outcome, with a survival rate of 100%, the absence of events in this subgroup limits the estimation of the risk using the Cox model.

#### 3.3.2. Relapses at the End of the Study (at 15 Months)

Subjects in both groups were followed up six months after completion of M-ECT in the M-Pharm/ECT group. In these six months, two subjects dropped out early and two relapsed, all four in the M-Pharm/ECT group. Nineteen subjects of the total sample (54.3%) had relapsed at the end of the 15 month follow-up: 47.1% (8/17) in the M-Pharm/ECT group and 61.1% (11/18) in the M-Pharm group, with no statistically significant differences between groups (*p* = 0.404). Among subjects who relapsed (n = 19), the median (min-max) time to relapse was 3.4 months (0.5–14.9): 4.2 months in the M-Pharm/ECT group (0.8–14.9) vs. 3 months in the M-Pharm group (0.5–6.6), with no statistically significant differences between the groups (*p* = 0.145). Table 4 describes disease-free survival at 3, 6, 9, 12, and 15 months (see Kaplan–Meier curves in Appendix A).

### 3.4. Changes in Clinical Scales (HDRS-21 and CGI-S)

No significant differences were found between the two treatment arms in HDRS-21 and CGI-S scores either at baseline or at the end of follow-up (Table 5). However, the HDRS-21 scores increased significantly between baseline and the end of follow-up in the global sample and in the M-Pharm group (though not in the M-Pharm/ECT arm, Table 5 and Figure 3). The CGI-S increased significantly between baseline and end of follow-up in the global sample and in both treatment groups (Table 5). The within-subjects (time and interaction between time and treatment arm) and between-subjects (treatment arm) factors were both significant in the GLM for HDRS-21 (*p* < 0.05, Figure 3) and CGI-S (*p* < 0.05).

### 3.5. Adverse Events and Cognitive Function

In the M-Pharm group, there were no early dropouts due to adverse drug effects. Similarly, in the M-Pharm/ECT group, no subjects dropped out early due to adverse effects related to ECT. One subject in the M-Pharm group was hospitalized for a suicide attempt.

The UKU subscale scores (psychological side effects, neurological side effects, autonomic side effects, and other side effects) were very low in both treatment groups. No significant differences were found between the two treatment arms in the UKU subscale scores, either at baseline or at the end of follow-up. A statistically significant increase was observed in the M-Pharm group with respect to the other side effects subscale score (*p* = 0.012).

At baseline, the MMSE scores did not differ significantly between the treatment arms (*p* = 0.512, Table 1). There were no statistically significant differences between the two arms at nine or 15 months (Table 6). The MMSE score for the M-Pharm/ECT group was similar throughout the study; in the M-Pharm group, there was a non-significant improvement at nine months, which remained stable throughout the study (Table 6).

## 4. Discussion

Our study evaluates the efficacy of extending ECT (plus pharmacotherapy, M-Pharm/ECT group) beyond six months (M-ECT for nine months) versus pharmacotherapy alone (M-Pharm) as maintenance strategies in severe MDD after achieving clinical remission with an acute course of ECT. Relapse rates were 35.3% in the M-Pharm/ECT group and 61.1% in the M-Pharm group, with no statistically significant differences between the treatment groups. Though not statistically significant, this percentage difference exceeding 25% between the two groups may be clinically relevant; indeed, the lack of statistical significance may be due to the relatively small sample size. However, although our forecast was to include 108 patients, the anticipated inclusion rate was too slow to prolong the study within available resources and the study had to be terminated before the planned number of patients was reached. No differences were found between treatment groups with respect to time to relapse, which was in line with previous studies [11,47], most relapses occurred at the beginning of maintenance treatment.

Residual symptoms worsened over the clinical course in the global sample and in both treatment groups, with a statistically significant increase in the M-Pharm group. These residual symptoms were more severe in the M-Pharm group from 2.5 months (Assessment 7) to the end of follow-up (15 months, Assessment 16). Finally, with regard to tolerability, both treatments were well tolerated with no significant differences between groups.

The relapse rate in the M-Pharm/ECT group was similar to those reported in other randomized studies [11,14]. The CORE group (Consortium for ECT) conducted a multisite, randomized, six-month trial to assess the efficacy of continuation electroconvulsive therapy (C-ECT) in MDD [11]. In the C-ECT group, 37.1% of subjects experienced disease relapse as compared with 31.6% of subjects in the C-Pharm group (lithium plus nortriptyline). No differences were detected between treatment groups. However, the authors concluded that C-ECT was an effective and safe strategy and superior to a placebo when comparing their results with those obtained by Sackeim et al. [47]. Nordenskjöld et al. found a significant advantage for M-ECT plus pharmacotherapy as compared with pharmacotherapy alone: 32% of the subjects treated with M-ECT plus pharmacotherapy versus 61% of those treated with pharmacotherapy relapsed within a year (*p* = 0.036) [14]. From this study, their best estimate was an absolute reduction in relapse rate of approximately 29%; thus, the number to treat for one subject to benefit would be around three to four [14], which was in the same range as ours. In another randomized trial, a significant advantage for combined treatment (ECT plus pharmacotherapy) was reported in a small sample of geriatric subjects with psychotic symptoms [13]. In the PRIDE Study (Prolonging Remission in Depressed Early, Phase 2), the relapse rate was lower than in other studies: only 17% of subjects relapsed [12]. However, the authors pointed out that the study was not powered to allow valid inferential results for comparing relapse between treatment groups (C-ECT and venlafaxine plus lithium vs. venlafaxine plus lithium). One possible reason for this lower relapse rate could be that the study protocol allowed the administration of additional or rescue C-ECT sessions. In the PRIDE Study, the C-ECT plus medication arm consisted of an initial fixed schedule (four right unilateral ultrabrief pulse ECT treatments in one month) followed by an individualized flexible schedule. In the rest of the studies, the ECT procedure involved only fixed sessions (fixed schedule): ten bilateral ECT sessions over six months (1.7 sessions per month) [11], 29 unilateral ultra-brief pulse ECT sessions over 12 months (2.4 sessions per month) [14], and 14 bilateral brief pulse ECT sessions over nine months (1.5 sessions per month) in our study.

Regarding the identification of potential contributing factors, only the absence of psychotic symptoms at the index episode seems to play a role in relapse. In the M-Pharm/ECT group, no subjects with psychotic symptoms relapsed in the first nine months. This finding is in line with previous studies suggesting that ECT is particularly effective in subjects with depression with psychotic features [13,48]. The meta-analysis of Van Diermen et al. (2018), for example, found the presence of psychotic features to be a predictor of ECT remission (odds ratio (OR) = 1.47, *p* = 0.001) and response (OR = 1.69, *p* < 0.001) in subjects with major depression [49].

Literature on the clinical outcome of subjects discontinuing M-ECT is limited, but the data available suggest high rates of relapse and recurrence ranging from 44% to 50%, especially in the first year after discontinuation [50,51]. In addition, neither the duration of M-ECT nor the criteria for its discontinuation have been clearly defined [1,3]. In our study, subjects were followed up for six months after discontinuation of M-ECT for a total of 15 months. Although no significant differences between treatment groups were found, it is worth noting that, during the period of active M-ECT, survival was higher in the M-Pharm/ECT group. When M-ECT was withdrawn, and especially six months after discontinuation, survival in both groups tended to even out, which may suggest that M-ECT has a protective effect on subjects with severe depression.

Regarding the outcome of the depressive symptoms, a significant increase in severity was observed at the end of follow-up. The study sample as a whole showed higher scores on the HDRS-21 at 15 months as compared with the baseline, although the difference was found to be significant only in the M-Pharm group. In this regard, in the PRIDE study [12], at 24 weeks the ECT plus medication group had significantly lower HDRS scores than the medication-only group; in addition, there were also significantly more subjects in the C-ECT plus medication group who were rated as “not ill at all” on the CGI-S [12]. In our study, the CGI-S increased significantly in both treatment groups at 15 months of follow-up.

Finally, M-ECT is a safe and well-tolerated treatment. In our study, both treatment strategies were well tolerated, with few adverse effects reported. Likewise, there was no statistically significant difference between groups in MMSE scores at baseline and at the end of study, in agreement with other trials [11,12,14].

This study has the following strengths and limitations. It applied a sound, randomized, prospective study design, but the sample size was small. As in similar previous studies, the recruitment of participants was particularly challenging; the selection of the 37 subjects finally randomized accounts for only 24% of the total of 152 subjects evaluated, a figure similar to the 28% (56 randomized/200 evaluated) in a study by Nordenskjöld et al. (2013) [14] and lower than in the studies by Kellner et al., which were 59% (201 randomized/341 screened) [11] and 50% (120 randomized/240 screened) [12]. This limitation might be overcome by the design of large-scale multicenter trials in order to increase the sample size, but these are more costly and difficult to standardize. The use of a fixed M-ECT guideline may also constitute a limitation; indeed, the lack of flexibility in the continuation ECT schedule was also reported as a potential limitation in the CORE study [11]. In contrast, the PRIDE study [12] opted to implement a more flexible guideline, the STABLE algorithm (Symptom-Titrated, Algorithm-Based Longitudinal ECT [52]). However, this algorithm has not demonstrated its superiority, and, due to its complexity, it is not widely used in real-world practice. The use of a flexible or individualized regimen depending on the subject’s requirements is now recommended, especially in the initial months of C/M-ECT therapy, in order to provide additional or rescue sessions when necessary [3,53]. Another limitation of the study is the heterogeneity of the pharmacotherapy given, even though this reflects a real-world practice. The low percentage of subjects receiving lithium was an interesting finding, perhaps due to the relatively advanced age of the sample that may have limited the use of this drug. Regarding other potential limitations, only bilateral ECT was prescribed; therefore, further studies should be performed in both bilateral and unilateral ECT, using distinct pulse widths. Finally, the non-normality of the data at baseline could also be a limitation for the GLM.

We consider that the lack of statistically significant differences may be mainly due to the low statistical power. Therefore, the absence of statistically significant differences should not be categorically interpreted as a lack of clinical relevance of M-ECT. Despite the limited statistical power, some potential results of our study, such as the outcomes in subjects with psychotic symptoms and in residual depressive symptoms, may be clinically significant and may support the use of M-ECT, especially in cases of severe illness.

## 5. Conclusions

In line with Kellner et al. [11,12], our data demonstrate moderate protection against depressive relapse using both long-term treatment options, but do not provide statistical evidence to suggest that one treatment option is more effective in preventing relapse than the other. Early intensive individual therapeutic interventions are needed in depressed subjects requiring ECT in the acute phase in order to prevent future relapses, as are further studies designed to identify predictors of relapse after recovery from an episode of major depression with acute ECT and to define the indications for M-ECT more precisely.

## Figures and Tables

**Figure 1 brainsci-11-01340-f001:**
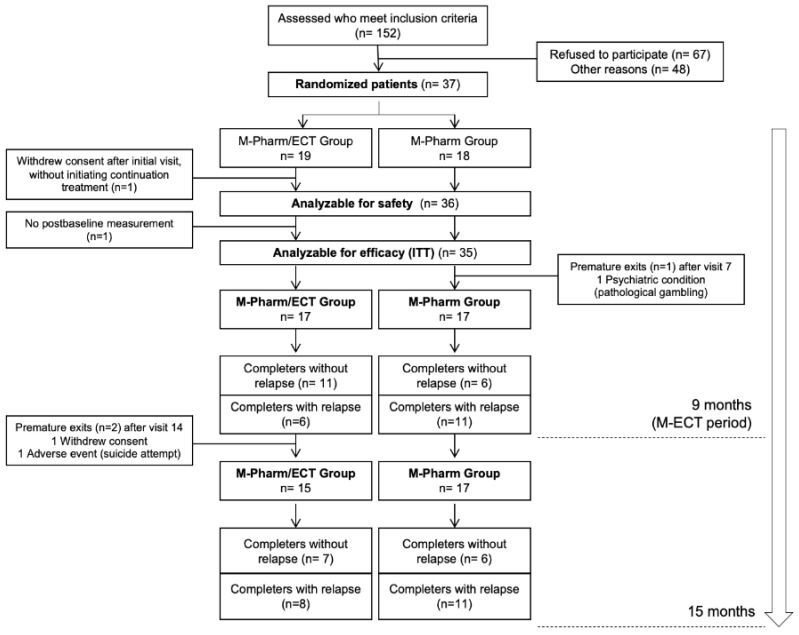
Subject flowchart.

**Figure 2 brainsci-11-01340-f002:**
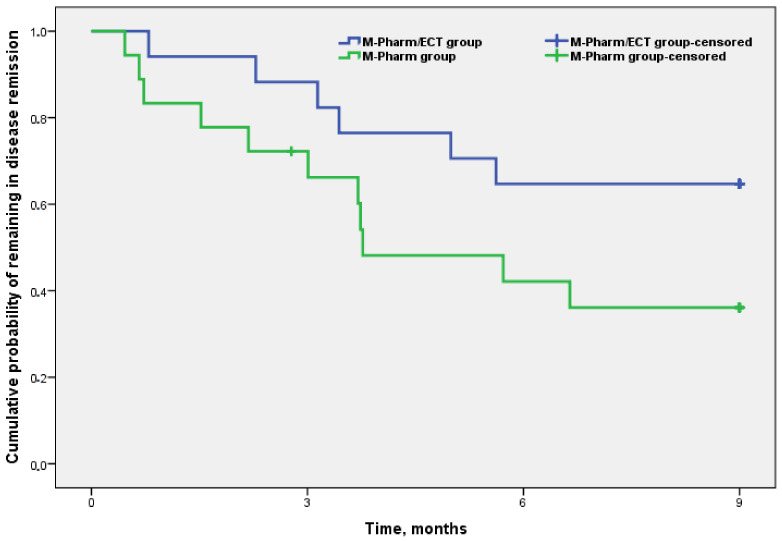
Kaplan–Meier function of the cumulative probability of remaining in remission at 9 months.

**Figure 3 brainsci-11-01340-f003:**
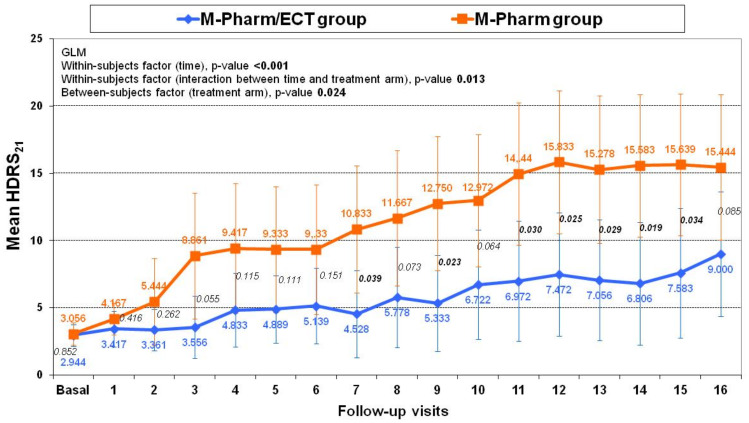
Changes in the HDRS-21 score between treatment groups (means, 95% CI).

**Table 1 brainsci-11-01340-t001:** Pharmacotherapy at baseline and during the study by treatment groups.

	M-Pharm/ECT Group n = 18	M-Pharm Group n = 18	*p*-Value
Serotonin selective reuptake inhibitor, % (n)	5.6 (1)	16.7 (3)	0.603 ^b^
Serotonin norepinephrine reuptake inhibitor, % (n)	77.8 (14)	83.3 (15)	1.000 ^b^
Noradrenergic and specific serotonergic antidepressant, % (n)	44.4 (8)	38.9 (7)	0.735 ^a^
Tricyclics, % (n)	33.3 (6)	27.8 (5)	0.717 ^a^
Other antidepressants, % (n)	22.2 (4)	11.1 (2)	0.658 ^b^
Lithium, % (n)	5.6 (1)	0 (0)	1.000 ^b^
Antipsychotics, % (n)	44.4 (8)	83.3 (15)	0.035 ^b^
Benzodiazepines, % (n)	77.8 (14)	61.1 (11)	0.471 ^b^

^a^*p*-value, χ^2^ test; ^b^
*p*-value, Fisher test.

**Table 2 brainsci-11-01340-t002:** Baseline characteristics of the subjects included (analyzable for safety sample).

	M-Pharm/ECT Group (n = 18)	M-Pharm Group (n = 18)	*p*-Value
**Demographic and course of the disorder characteristics**			
Sex (female), % (n)	72.2 (13)	61.1 (11)	0.480 ^a^
Age, median (min-max), years	69 (34–81)	67 (52–77)	0.889 ^d^
Age at illness onset *, mean (SD), years	49.9 (15.2)	50.8 (18)	0.877 ^c^
Recurrent MDD, % (n)	94.4 (17)	66.7 (12)	0.088 ^a^
Prior episodes, median (min-max)	3.0 (1–10)	3.0 (1–12)	0.591 ^d^
Use of ECT in previous episodes, % (n)	38.9 (7)	27.8 (5)	0.480 ^b^
**Characteristics of the index episode**			
Melancholic symptoms, % (n)	94.4 (17)	100 (18)	1.000 ^b^
Psychotic symptoms, % (n)	44.5 (8)	72.3 (13)	1.000 ^b^
Catatonic symptoms, % (n)	11.1 (2)	5.6 (1)	1.000 ^b^
Duration of current episode, median (min-max), years	0.4 (0.1–2.7)	0.5 (0.1–4.1)	0.527 ^d^
Treatment-resistant depression, % (n)	55.6 (10)	33.3 (6)	0.180 ^b^
Acute ECT sessions, median (min-max)	10.5 (7.0–17.0)	12.0 (7.0–21.0)	0.191 ^d^
**Characteristics at baseline**			
HDRS_21_, median (min-max)	3.0 (0.0–6.0)	2.0 (0.0–6.0)	0.834 ^d^
CGI-S, median (min-max)	1 (1–2)	1 (1–2)	0.971 ^d^
CORE system, median (min-max)	1.0 (0.0–4.0)	1.0 (0.0–6.0)	0.268 ^d^
GAF, median (min-max)	80 (70–90)	80 (70–90)	0.739 ^d^
MMSE, mean (SD)	25.67 (3.87)	24.75 (4.19)	0.512 ^c^

MDD, major depressive disorder; ECT, electroconvulsive therapy; HDRS-21_,_ Hamilton Depression Rating Scale (21 items); CGI-S, Clinical Global Impression Severity scale; GAF, Global Assessment of Function; MMSE, Mini Mental State Examination. * Missing data for one subject. ^a^
*p*-value, χ^2^ test; ^b^
*p*-value, Fisher test; ^c^
*p*-value, Student’s *t*-test; ^d^
*p*-value, Mann–Whitney U.

**Table 3 brainsci-11-01340-t003:** Survival analysis results from Cox proportional hazards regression models at 9 months.

Model	Hazard Ratio (95% CI)	*p*-Value	Hazard Ratio (95% CI)	*p*-Value *
**Model 1** Treatment (Pharm)	2.3 (0.85–6.23)	0.102		
**Model 2** Treatment (Pharm) Psychosis (without) Age (>67 years)	2.79 (0.998–7.771) 3.57 (1.226–10.382) 0.44 (0.146–1.306)	0.050 **0.020** 0.138	3.17 (1.141–8.781) 4.56 (1.622–12.822) - - - -	0.027 **0.004** - - - -

Model 1 contained treatment, and Model 2 (adjusted model) contained treatment, psychosis, and age. * Backward stepwise process (excluding from the multivariate model the variables with statistical significance > 0.05).

**Table 4 brainsci-11-01340-t004:** Cumulative probability (% (95% CI)) of remaining in disease remission at 3, 6, and 9 months (active treatment with M-ECT in M-Pharm/ECT group) and at 12 and 15 months (M-ECT discontinued) by treatment groups.

	Active M-ECT			Stop M-ECT *	
	3 Months	6 Months	9 Months	12 Months	15 Months
**M-Pharm/ECT group**	88.2 (72.9–103.5)	64.7 (42.0–87.4)	64.7 (42.0–87.4)	57.5 (33.3–81.7)	38.3 (3.6–73)
**M-Pharm group**	72.2 (51.5–92.9)	42.1 (18.6–65.6)	36.1 (13.2–59.0)	36.1 (13.2–59)	36.1 (13.2–59)

Log-rank test = 0.093 at 9 months and 0.143 at 15 months. * Pharmacotherapy only in both groups.

**Table 5 brainsci-11-01340-t005:** Differences between baseline vs. end of follow-up (15 months) in the HDRS-21 and CGI-S scales in the ITT sample.

	M-Pharm/ECT Group (n = 17)	M-Pharm Group (n = 18)	*p*-Value
	Mean (SD); Median (Min-Max)	Mean (SD); Median (Min-Max)	
**HDRS-21 at baseline**	**3.1 (1.6); 3.0 (0.5–5.5)**	**3.1 (1.8); 3.0 (0.0–6.0)**	**0.915 ^a^**
**HDRS-21 at 15 months**	**9.5 (10.1); 3.0 (0.0–25.0)**	**15.4 (11.7); 17.5 (0.0–35.0)**	**0.209 ^b^**
***p*-Value ***	**0.125 ^d^**	< 0.001 ^c^	
**CGI-S at baseline**	**1.4 (0.4); 1.0 (1.0–2.0)**	**1.4 (0.5); 1.0 (1.0–2.0)**	**0.926 ^b^**
**CGI-S at 15 months**	**2.5 (1.6); 1.5 (1.0–5.0)**	**3.4 (1.8); 4.0 (1.0–6.0)**	**0.171 ^b^**
***p*-Value ***	0.016 ^d^	0.001 ^d^	

^a^*p*-value, Student’s *t*-test; ^b^*p*-value, Mann–Whitney U test; ^c^*p*-value, repeated measures *t*-test; ^d^*p*-value, Wilcoxon test. *p*-value *, baseline at 15 months.

**Table 6 brainsci-11-01340-t006:** Development of measures of cognitive function by MMSE among subjects who had not relapsed or dropped out (n = 13, missing data for one subject).

	M-Pharm/ECT Group (n = 5)	M-Pharm Group (n = 7)	*p*-Value
**MMSE at baseline, mean (SD)**	**25.00 (5.1)**	**24.20 (3.96)**	**0.776 ^a^**
**MMSE at 9 months, mean (SD)**	**24.57 (5.35)**	**27.40 (1.14)**	**0.277 ^a^**
**MMSE at 15 months, mean (SD)**	**25.14 (4.34)**	**27.80 (1.64)**	**0.226 ^a^**
***p*-Value ^1^** ***p*-Value ^2^**	**0.482 ^b^**	**0.145 ^b^**	
	**0.924 ^b^**	**0.125 ^b^**	

^a^*p*-value, Student’s *t*-test; ^b^*p*-value, repeated measures *t*-test. *p*-value ^1^, baseline at 9 months; *p*-value ^2^, baseline at 15 months.

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
