# Peer review of "Can the Addition of Maintenance Electroconvulsive Therapy to Pharmacotherapy Improve Relapse Prevention in Severe Major Depressive Disorder? A Randomized Controlled Trial"

_brainsci, 2021, doi:10.3390/brainsci11101340_

Round 1
Reviewer 1 Report
I am happy with the revisions. The paper can now be accepted. Congratulations to the authors.
Reviewer 2 Report
Thank the authors for addressing my concerns.
This manuscript is a resubmission of an earlier submission. The following is a list of the peer review reports and author responses from that submission.
Round 1
Reviewer 1 Report
The authors present a randomized, single-blind trial of maintenance ECT with pharmacotherapy versus pharmacotherapy alone for severe major depression. This study found no significant difference between the arms of the study in terms of efficacy, tolerance, dropout rate, and relapse rate, at the end of the trial and at follow-up. However, they have found a significant moderating factor, that is the presence of psychotic symptoms. Additionally, the authors posited that the low power/ sample size may have played a role in the study results.
The manuscript is well written and supported by several tables and figures. Overall, however, it misses on several points that need rectification:
- They have not looked for the effect of moderation factors such as seizure duration, electrical dose, and anesthetic medication, which could have provided some direction for future studies. For example, a recent meta-analysis has shown an investigation of the various ECT modalities to help with treatment response examination [See Ainsworth et al 2020, J ECT 36(2):94-105].
- There is no mention as to whether the psychiatrist(s) managing medications was/were blinded to ECT groupings.
- The use of Bonferroni correction at onset was not appropriate and should only have been used on end-points outcomes. Hence, the result at onset, suggests that the groups were not balanced based on antipsychotics drug usage and suggesting symptom severity differences; the ECT group had a lower % of antipsychotic use at the onset! This may explain the finding of psychosis as a significant moderating factor.
- On page 4, between lines 143 and 165, information about MMSE, CGI, and HDRS seems to have redundancy. This paragraph/section needs better writing.
- On page 4, line 165, the authors suggested that they have used 15-17 and 18 as cut-off scores, why so many cut-offs? This may have affected patient classification and the overall results.
- On page 4, line 157, the author reports that other neurocognitive tests were done and reported elsewhere, but no reference provided.
- On page 5, line 196, the authors suggest that they have used the p-value of 0.05 as a cut-off for significance; how is it that here they did not correct for the number of variables and analysis and mentioned the Bonferroni?
- As per table 2, most of the results reported are in median (min-max) with the exception of the MMSE scores. How was this variation and skewness in the data was taken into consideration in the regression analysis? Was there a significant difference among subjects within groups? Or can we assume that the samples within each arm were heterogeneous to start with?
- The authors report on the global sample, why does it matter, when we are comparing two groups?
- The authors report on one subject in the M-pharm group to have been hospitalized for a suicide attempt, was this an adverse event or psychological side-effect? If any of the above, then this may affect the non-significant difference found on line 343, page 10.
Reviewer 2 Report
This is a single-blind multicenter study that establishes the efficacy of combinatorial treatment of pharmacological agents and ECT when compared to pharmacological treatments alone in MDD. The manuscript is very well written and describes all the data clearly and coherently. I have very few comments.
One of the main drawbacks of this study is the low subject numbers. What do the author infer from this data considering such low subject numbers. It would be nice to clearly delineate that in the discussion.
One of the secondary aims the authors describe here is tolerability. what are the different parameters on which the tolerance was determined? The text just mentions that the treatments were well tolerated.
The heterogeneity in the drug groups also makes the interpretations difficult. It is not clear whether any of these drugs were discontinued or started at a later time/resumed? Did the authors collect those data?
As it appears this is a study that included exclusively females. Due to the median age of the subjects, it is hard to determine if all of them had undergone menopause or not. It is also not clear if any of these subjects were on any form of hormone therapy.
Line 382: Please correct the extra space.
Line 232: As regards clinical characteristics, 97.2%: a to is missing after regards.
Line 27: There were no statistically significant differences in
the MMSE scores of the two groups at the end of study: Please add "the" before study.
Lines 37 and 38: In fact, nearly 40% of ECT responders can be expected to relapse in the first six months, and roughly 50% by 38 the end of first year: Please rephrase. It does not read well.
Line 193: Student's T test or repeated measures t-test, as appropriate. Please maintain consistency when designating t-test.
Reviewer 3 Report
In this manuscript, the authors reported the results of an RCT investigating the effectiveness of addition of maintenance ECT to pharmacotherapy in preventing the relapse of severe MDD. This is an interesting and useful topic, and the manuscript is generally well-written and the results clearly presented. However, a fatal issue with this manuscript is its small sample size; as an RCT, the authors should have conducted a priori power analysis to determine the minimum sample size required to detect how big an effect size. Without such information, it is hard to know if the nonsignificant finding regarding the primary outcome of the current study is genuine or just that the study is underpowered to detect a significant effect.